# Mediating Effects of Rehabilitation Motivation between Social Support and Health-Related Quality of Life among Patients with Stroke

**DOI:** 10.3390/ijerph192215274

**Published:** 2022-11-18

**Authors:** Yaeram Lee, Mihwa Won

**Affiliations:** Department of Nursing, Wonkwang University, Iksan 54538, Republic of Korea

**Keywords:** social support, rehabilitation, motivation, quality of life

## Abstract

Post-stroke health-related quality of life (HRQoL) is poor, which is related to low social support levels and lack of rehabilitation motivation. However, there are limited studies that have systematically analyzed the mechanisms underlying this relationship in stroke patients. This study aimed to identify the mediating effects of rehabilitation motivation on the relationship between social support and HRQoL among stroke patients. A cross-sectional descriptive study was conducted on 176 Koreans aged ≥19 years who were admitted for rehabilitation treatment after stroke at three general hospitals in Jeonbuk. Data collection was conducted from September to December 2020 through face-to-face interviews using structured questionnaires and review of medical records. The significance of the mediation model was tested using SPSS 25.0 and the PROCESS macro for SPSS v3.5. Significant associations were identified between social support and HRQoL. Our findings revealed an indirect effect, suggesting that the effects of social support on HRQoL are mediated by rehabilitation motivation (B = 0.004, 95% bias-corrected bootstrap confidence interval = 0.002, 0.006). Social support for stroke patients had a positive effect on HRQoL, and rehabilitation motivation was found to have a partial mediating effect on this relationship. This study suggests that social support from healthcare professionals and families for post-stroke can improve patients’ HRQoL by inducing positive rehabilitation motivation. Therefore, developing intervention strategies to motivate rehabilitation could improve the HRQoL of patients with stroke.

## 1. Introduction

An aging population and an increase in chronic diseases have increased the prevalence of stroke worldwide [1]. In the European Union, the prevalence of stroke is expected to increase by approximately 34% between 2015 and 2035 [2]. The prevalence of stroke in Korea was 1.71% in 2018, with an estimated 105,000 new stroke cases occurring annually [3,4]. Stroke with high morbidity and recurrence rates requires long-term disease management at both individual and national levels, which reduces the health-related quality of life (HRQoL) of patients along with increasing the economic burden of the disease [2,4].

Generally, stroke patients have a high incidence of disability after onset [1]. Surviving stroke patients experience various physical disabilities, such as motor and sensory impairments, speech and cognitive impairments, and gait disturbances due to neurological damage [5]. These symptoms of disability cause difficulties in the patient’s independent daily life performance, limit social activities and roles, and affect not only psychological disorders, such as anxiety and depression, but also HRQoL [6,7,8]. 

Considering the importance of HRQoL among patients with stroke, American Heart Association and American Stroke Association guideline aim to prevent complications in adult patients after stroke as a clinical treatment goal, minimize physical disabilities, and promote HRQoL [5]. A previous study has found that the level of HRQoL of Australian stroke patients with speech impairment and hemiplegia in acute hospitals is lower than that of general adults [9]. Also, a meta-analysis of stroke rehabilitation reported that the level of physical or psychological quality of life decreases according to the rehabilitation stage among patients with stroke [10]. Moreover, a longitudinal cohort study of stroke patients enrolled in a South African rehabilitation center has shown that post stroke HRQoL was associated with stroke severity and function [11]. Thus, early assessment of the HRQoL of stroke patients in the clinical field is useful for assessing multidimensional health status, such as an individual’s physical, psychological, emotional, and social functions, and is vital to understanding treatment effects, such as readmission and mortality [11,12]. 

Factors influencing HRQoL of patients with stroke include sociodemographic factors, such as gender, age, education level, hospitalization period, and economic condition; clinical factors, such as physical functional status, disease duration, comorbidities, and rehabilitation period; and social factors, such as support, confidence, and motivation for rehabilitation among patients with stroke [10,11,13,14,15]. Particularly, social support is a proven coping mechanism for emotional suffering and has been reported to affect medical outcomes among patients with chronic diseases [16]. 

Social support has been reported in previous studies as a predictive factor that promotes functional recovery by reducing negative emotional responses such as stress and helping to adapt to a changed life after stroke [15,16]. In addition, there is a significant positive correlation between social support and HRQoL among patients with stroke [16], and stroke patients with a high level of social support are correlated with the level of independent daily life performance [17]. A meta-analysis study reported that social support was found to be a powerful predictor of HRQoL among patients with stroke after adjusting for demographic and sociological factors [18]. Moreover, individuals with high perceived social support have a high degree of interest in and encouragement from healthcare professions and their families [17,19]. Recent studies have shown that individuals with abundant social support are more motivated to participate in rehabilitation [13,19]. Thus, continuous rehabilitation treatment promotes HRQoL among stroke patients experiencing physical disability symptoms and improves stroke treatment results [12,14,20]. 

A previous study found that individuals with higher rehabilitation motivation had higher physical and mental quality of life [21]. In a recent study, it was reported that the rehabilitation motivation was a key predictor of HRQoL among stroke inpatients [22]. Moreover, some previous studies have reported that the HRQoL of stroke patients increases after applying the motivational nursing intervention program [23,24]. Hence, stroke patients should be active participants in rehabilitation; in this regard, it is important to promote motivation for rehabilitation [22,24]. Consequently, for successful rehabilitation in patients with stroke, it is necessary to pay attention to rehabilitation motivation, which is a key element of change [13]. 

Rehabilitation motivation is induced by the social support of health providers or family members during the rehabilitation process, which has a positive effect on HRQoL of stroke patients [23]. In particular, it was reported that social support considering comprehensive information on individuals’ general factors, health status, and environmental factors by healthcare professionals increases adherence to rehabilitation among stroke patients [21]. Consequently, social support is associated with rehabilitation motivation and improvement in HRQoL among patients with stroke [25,26]. 

Although some studies have been conducted to analyze correlations between only two factors of each factor, such as social support, rehabilitation motivation, and HRQoL, few have investigated the association of all three factors among patients with stroke. In this regard, understanding of the concrete mechanism for the mediating effect of rehabilitation motivation in the relationship between social support and HRQoL among patients with stroke is limited. Thus, it is necessary to investigate the mediating effect of rehabilitation motivation in the relationship between social support and HRQoL among patients with stroke, which would help develop rehabilitation motivation strategy to improve HRQoL among patients with stroke.

## 2. Materials and Methods

### 2.1. Aimes

This study aimed to confirm the relationship between social support, rehabilitation motivation, and HRQoL among patients with stroke and to identify the mediating effect of rehabilitation motivation on the relationship between social support and HRQoL. The three hypotheses of this study were formulated.

 **Hypothesis 1 (H1).**
*Social support is associated with HRQoL among patients with stroke.*


 **Hypothesis 2 (H2).**
*Social support is associated with rehabilitation motivation among patients with stroke.*


 **Hypothesis 3 (H3).**
*Rehabilitation motivation is associated with HRQoL among patients with stroke.*


 **Hypothesis 4 (H4).**
*Rehabilitation motivation plays mediating role in the association between social support and HRQoL among patients with stroke.*


### 2.2. Study Design and Sample

A convenience sample was obtained from stroke patients admitted to the rehabilitation departments of three general hospitals in Jeonbuk, Korea. The inclusion criteria were as follows: (a) age 19 years or older; (b) admitted to the rehabilitation ward after a stroke diagnosis; (c) over 24 points on the Korean version of the Mini-Mental State Examination [26]; and (d) ability to read and write the Korean. 

The exclusion criteria were as follows: (a) diagnosed with Wernicke’s aphasia by a physician; (b); being classified in the modified Rankin Scale (mRS) class severe disability (>5 scores) [27], which requires continuous care; and (c) diagnosis history of dementia, depression, severe end-stage cancer, and chronic kidney disease. 

We performed a power analysis to determine the appropriate sample size using computer program G-power version 3.1.9 [28]. Based on a significance level (α) of 0.05, power of 0.90, moderate effect size (0.15), and 13 predictors, the calculation revealed that 162 participants were required for detection. A total of 190 questionnaires were distributed to account for the dropout rate. After excluding insincere responses and non-response questionnaires, 176 questionnaires (response rate: 92.6%) were used for the final analysis.

### 2.3. Data Collection

The data collection of stroke patients was conducted after obtaining prior permission at three general hospitals from September to December 2020. A poster was attached to the bulletin board of each hospital’s rehabilitation department ward, and three rehabilitation doctors confirmed that inpatients who voluntarily participated met the inclusion criteria. Participants completed the questionnaires in the consultation room, and the survey took approximately 20 min to complete. The clinical characteristics were confirmed using an electronic medical chart. 

### 2.4. Ethical Considerations

This study protocol was approved by the Institutional Research Board of Wonkwang University, Jeonbuk, South Korea. Before filling out the questionnaire, the researcher explained the purpose of the study to all participants to consider the ethical rights of the participants. The researcher ensured the autonomy, confidentiality, and freedom to withdraw to all participants who voluntarily participated in the study.

### 2.5. Instruments

#### 2.5.1. Sociodemographic Characteristics and Disease-Specific Characteristics of Stroke Patients

Sociodemographic characteristics and disease-specific characteristics of stroke patients were collected using a self-reported questionnaire, including age, gender, educational level, family type, job status, monthly income, body mass index (BMI), time since diagnosis, mRS score, and paralytic region, such as left paralysis, right paralysis, and speech impediment. Body mass index was used by dividing the participant’s weight by the square of their height. According to the guidelines of the World Health Organization for the Asia-Pacific region and the obesity standards set by the Korean Society for Obesity, underweight is below 18.5 kg/m^2^, normal weight is 18.5 to 22.9 kg/m^2^, overweight is 23 to 24.9 kg/m^2^, and obese is above 25 kg/m^2^. The mRS is classified by a 0–6 score, and it is a tool to measure the degree of disability or dependence on daily activities of stroke patients [27]. 

#### 2.5.2. Social Support

Social support was measured using a social support scale with proven reliability and validity to assess the social support from healthcare professionals and families of stroke patients [29]. This measurement consists of total 16 items and the 5-point Likert scale ranges from 1 to 5: the higher the score, the higher the social support healthcare professionals, and families. Cronbach’s α was 0.87 at the time of development and 0.90 in this study.

#### 2.5.3. Rehabilitation Motivation

The Stroke Rehabilitation Motivation Scale (SRMS) was used to assess the reactivity period of patients with stroke [30]. In our study, the rehabilitation motivation of stroke patients was measured using the validated Korean version of the Stroke Rehabilitation Motivation Scale [31]. This scale consists of 24 items and is scored on a 5-point Likert scale ranging from 1 to 5. A higher score indicates a higher motivation for rehabilitation. Cronbach’s α was 0.70 for the Korean version of the SRMS. The reliability value in our study was Cronbach’s α = 0.86.

#### 2.5.4. Health-Related Quality of Life

The European Quality of Life Scale-5 Dimension (EQ-5D) index assesses an individual’s standardized multidimensional HRQoL [13]. After registering for use of EQ-5D, the Korean version of the EQ-5D with established reliability and validity was used in this study [13]. This tool consists of 5 multidimensions, including mobility, daily activities, self-care, pain/discomfort, and anxiety/depression. HRQoL was comprehensively presented by applying Korean weights for the five-dimensional health status of the EQ-5D to calculate one indicator score between 0 (theoretical death) and 1 (full health) based on the mapping method proposed by the EuroQol group. Each item is divided into three scales: ‘Level 1′ for no problems, ‘Level 2′ for moderate problems, and ‘Level 3′ for severe problems. The score is calculated as 1 point if the answer is no problem in all five areas.

### 2.6. Statistical Analysis

The collected data were analyzed using SPSS 26.0 (IBM, Armonk, NY, USA). Normality of data was checked using skewness and kurtosis. Frequencies with percentages and means with standard deviations were used for sociodemographic characteristics and disease-specific characteristics of the stroke patients. Descriptive statistics of the mean with standard deviation were used for social support, rehabilitation motivation, and HRQoL among patients with stroke. Differences in HRQoL according to sociodemographic characteristics and disease-specific characteristics of stroke patients were analyzed by *t*-test and ANOVA, and the main correlation variables were analyzed using Pearson’s correlation coefficient. The significance of the mediating effect was analyzed using Hayes’ (2017) SPSS PROCESS macro model no. 4 [32]. The significance test of the indirect effect was performed using bootstrapping of the PROCESS macro with 10,000 samples and a confidence interval of 95.0%. Hayes’s (2017) process macro uses bootstrapping, which does not assume normality of the distribution of indirect effects, so it is a method to overcome the limitation of not reflecting the actual sampling distribution according to the assumption of normality of the Sobel test [32]. Therefore, Hayes’s (2017) process macro method is an analytical method with a relatively high power compared to the Sobel test.

## 3. Results

### 3.1. Sociodemographic Characteristics and Disease-Specific Characteristics of Stroke Patients 

The mean (SD) age of a total of 176 Korean stroke patients was 63.86 (SD 12.25) years, and 53.4% were below 65 years. More than half of the participants were men (60.8%), had above high school education level (57.4%), lived with family (72.2%), were unemployed (68.2%), and had less than 1 million won in monthly income. The mean (SD) body mass index was 23.31 (2.82) Kg/m^2^, and normal weight was most common (44.8%). Of the participants, 66.9% had a diagnosis time of >1 year. Regarding the mRS score, 50.6% scored 2 points. According to the paralysis region of the participants, right paralysis had the highest distribution (48.9%) (Table 1).

### 3.2. Differences in HRQoL According to the Characteristics of Stroke Patients

HRQoL differed significantly according to age (t = 2.53, *p* = 0.012), educational level (t = −2.13, *p* = 0.035), job status (t = 2.94, *p* = 0.004), time since diagnosis (t = 3.05, *p* = 0.003), and mRS score (t = 18.02, *p* < 0.001). However, there was no statistical significance in gender, family type, monthly income, BMI, and paralytic region such as left paralysis, right paralysis, and speech impediment (Table 1).

### 3.3. Correlation between Social Support, Rehabilitation Motivation, and HRQoL

The participants’ mean scores on social support, rehabilitation motivation, and HRQoL were 64.52 (SD = 11.57), 89.88 (SD = 11.39), and 0.60 (SD = 0.25), respectively (Table 2).

HRQoL was significantly positively correlated with social support (r = 0.42, *p* < 0.001), rehabilitation motivation (r = 0.51, *p* < 0.001), and reactivity. A significant positive correlation was found between social support and rehabilitation motivation (r = 0.59, *p* < 0.001) (Table 2). 

### 3.4. Mediation of Rehabilitation Motivation in the Relationship between Social Support and Health-Related Quality of Life

The results of the mediation analysis are presented in Table 3 and Figure 1. The total effect of social support on HRQoL was significant (H1 path: B = 0.004, *p* < 0.001, 95% CI = 0.001, 0.007), after adjusting for age (1 = below 65), education level (1 = below middle school), job (1 = none), duration of CVA diagnosis (1 = 1 year), and mRS (1 = 1). 

Also, the direct effects of social support on rehabilitation motivation (H2 path: B = 0.573, *p* < 0.001, 95% CI = 0.454, 0.691) and rehabilitation motivation on HRQoL (H3 path: B = 0.007, *p* < 0.001, 95% CI = 0.004, 0.010) were significant after adjusting for covariates. 

Moreover, the direct effect of social support on HRQoL was significant after adjusting for rehabilitation motivation and other covariates, (H4 path: B = 0.008, *p* < 0.001, 95% CI = 0.006, 0.011), indicating that rehabilitation motivation mediated the association between social support and HRQoL. 

Furthermore, the indirect effect of rehabilitation motivation was statistically significant (H2 × H3: B = 0.004, 95% bias-corrected bootstrap CI = 0.002, 0.006), suggesting that social support has an indirect positive effect on HRQoL via rehabilitation motivation.

## 4. Discussion

This study demonstrated that abundant social support was significantly positively related to enhanced HRQoL among stroke patients. In other words, stroke patients with high social support had high satisfaction with HRQoL. Some researchers have reported that social support, such as social connectivity, trust, interaction, and cohesion between health information providers and families, has a positive effect on HRQoL among stroke patients [33,34]. This finding is in line with a previous study that examined the relationship between social support and HRQoL [35]. 

Moreover, these results suggest that stroke patients with higher social support may enhance HRQoL by receiving more positive and hopeful support through broader social interaction with healthcare professions and families [19,21]. Thus, perceived social support, such as interest and encouragement of healthcare providers and family during the rehabilitation process, can motivate stroke patients for rehabilitation treatment and, consequently, improve HRQoL. 

In our study, social support was significantly positively associated with rehabilitation motivation. This finding is like those reported in prior studies [13,36]. Some researchers reported that stroke patients with high levels of social support had high rehabilitation motivation [20,37,38], and social support from family was found to have a positive effect on rehabilitation motivation [38]. Specifically, stroke patients may be motivated to continue participating in rehabilitation treatment while receiving support from healthcare professions for their goals and experiences of success and failure of rehabilitation treatment [19,24]. 

Interestingly, we found that rehabilitation motivation partially mediated the effect of social support on HRQoL among patients with stroke. In stroke patients with higher social support, HRQoL was found to be improved in patients with higher rehabilitation motivation than those with lower rehabilitation motivation. This indicates the mediating role of rehabilitation motivation on the relationship between social support and HRQoL among patients with stroke. These results support previous findings that stroke patients with high social support are more motivated to participate in rehabilitation [23,36], and there is a significant correlation between perceived social support and rehabilitation motivation in patients with stroke [39].

Furthermore, the fact that rehabilitation motivation partially mediated the relationship between social support and HRQoL involves the concept that strengthening social support for stroke patients undergoing rehabilitation can motivate them to actively participate in rehabilitation treatment, thereby improving HRQoL. Thus, healthcare professionals should recognize that practical psychosocial strategies based on social support can help improve the HRQoL of stroke patients by fostering their intrinsic or extrinsic motivation for rehabilitation treatment [40]. Future research should focus on developing an intervention strategy aimed at motivating stroke patients to participate in long-term rehabilitation to improve HRQoL and evaluating its effectiveness.

This study is meaningful in that it confirmed the partial mediating role of the rehabilitation motivation in the relationship between social support and HRQoL among patients with stroke, and suggested the importance of developing a rehabilitation motivation strategy to improve HRQoL among patients with stroke. Nevertheless, this study has several limitations. First, the participants of this study were patients undergoing rehabilitation at three general hospitals, but generalizing is difficult or impossible because the proportion of stroke patients with mRS scores below 2 was high. Second, the cross-sectional study design did not explain the causal relationships between social support, rehabilitation motivation, and HRQoL. Finally, our study used a self-administered questionnaire, which may be affected by response bias.

## 5. Conclusions

Social support for stroke patients had a positive effect on HRQoL, and rehabilitation motivation was found to have a partial mediating effect on this relationship. These results indicate that social support from healthcare professionals and families for stroke patients can improve patients’ HRQoL by inducing positive rehabilitation motivation. Thus, early support intervention to increase rehabilitation motivation seems to be necessary for improving the HRQoL and reducing the risk factors of stroke. 

## Figures and Tables

**Figure 1 ijerph-19-15274-f001:**
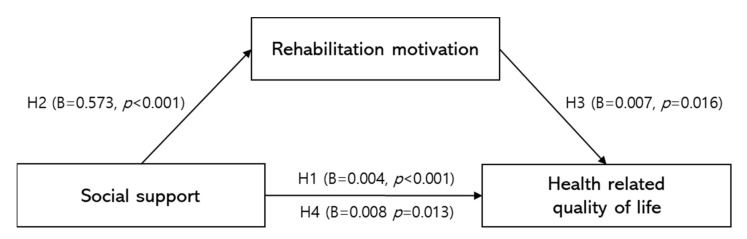
Mediating effect of rehabilitation motivation in patient with stroke. Adjusted for age (1 = below 65), education level (1 = below middle school), job (1 = no), duration of CVA diagnosis (1 =< 1 year), stroke severity (1 = 1).

**Table 1 ijerph-19-15274-t001:** Patients’ sociodemographic and clinical characteristics and differences in health-related quality of life (*n* = 176).

Characteristics	Category	*n* (%)	Mean (SD)	t or F	*p*
Age (years)	<65	94 (53.4)	0.64 (0.25)	2.53	0.012
	≥65	82 (46.6)	0.55 (0.23)		
Gender	Men	107 (60.8)	0.61 (0.26)	0.85	0.398
	Women	69 (39.2)	0.58 (0.23)		
Educational level	≤Middle school	75 (42.6)	0.56 (0.22)	−2.13	0.035
	≥High school	101 (57.4)	0.63 (0.26)		
Family type	Alone	49 (27.8)	0.57 (0.24)	−0.93	0.356
	Family	127 (72.2)	0.61 (0.25)		
Job status	Unemployed	120 (68.2)	0.57 (0.26)	2.94	0.004
	Employed	56 (31.8)	0.68 (0.19)		
Monthly income	<100	112 (63.6)	0.58 (0.26)	−0.38	0.704
(10,000 won)	≥100	23 (36.4)	0.60 (0.23)		
BMI (Kg/m^2^)	Underweight	6 (3.4)	0.58 (0.16)	1.19	0.313
	Normal	130 (73.9)	0.61 (0.25)		
	Overweight	34 (19.3)	0.55 (0.27)		
	Obese	6 (3.4)	0.74 (0.17)		
Time since diagnosis	<1	53 (30.1)	0.69 (0.21)	3.05	0.003
(years)	≥1	123 (69.9)	0.56 (0.25)		
mRS score	1	35 (19.9)	0.70 (0.26)	18.02	<0.001
	2	89 (50.6)	0.65 (0.13)		
	3–4	52 (29.5)	0.44 (0.31)		
Paralytic region					
Left paralysis *	No	96 (54.5)	0.57 (0.27)	−1.52	0.120
	Yes	80 (45.5)	0.63 (0.23)		
Right paralysis *	No	90 (51.1)	0.59 (0.26)	−0.54	0.590
	Yes	86 (48.9)	0.61 (0.23)		
Speech impediment *	No	138 (78.4)	0.59 (0.25)	−0.38	0.707
	Yes	38 (21.6)	0.61 (0.24)		

SD = standard deviation; BMI = body mass index; mRS = modified Rankin Scale. * Multiple response.

**Table 2 ijerph-19-15274-t002:** Correlations between social support, rehabilitation motivation, and health-related quality of life (*n* = 176).

Variables	Mean (SD)	Correlation Coefficients	
		1	2
		r (*p*)	r (*p*)
1. Social support	64.52 (11.57)	1	
2. Rehabilitation motivation	89.88 (11.39)	0.59 (<0.001)	1
3. Health-related quality of life	0.60 (0.25)	0.42 (<0.001)	0.51 (<0.001)

SD = standard deviation.

**Table 3 ijerph-19-15274-t003:** Mediating effect of rehabilitation motivation on the relationship between social support and health-related quality of life (*n* = 174).

Path	B	SE	t	*p*	95% CI		R^2^	F (*p*)
					LLCI	ULCI		
Social support → Rehabilitation motivation (H2 path)	0.573	0.060	9.53	<0.001	0.454	0.691	0.38	20.73 (<0.001)
Social support → HRQoL (H1 path)	0.004	0.002	2.80	0.005	0.001	0.007	0.42	20.93 (<0.001)
Rehabilitation Motivation → HRQoL (H3 path)	0.007	0.001	4.21	<0.001	0.004	0.010		
Social support → HRQoL (H4 path)	0.008	0.001	6.26	<0.001	0.006	0.011	0.37	19.62 (<0.001)
Indirect effect	coefficient = 0.004, bias-corrected bootstrap SE = 0.001, 95% bias-corrected bootstrap CI (0.002, 0.006)							

B: unstandardized coefficients; SE: standard error; LLCI: lower level of confidence interval; ULCI: upper level of confidence interval; HRQoL: health-related quality of life. Adjusted for age (1 = below 65), education level (1 = below middle school), job (1 = no), duration of CVA diagnosis (1 =< 1 year), and stroke severity (1 = 1).

## Data Availability

Data sharing is not applicable to this article as data analyzed in the current study contains participant medical information. Sharing of this information would breach patient privacy, confidentiality and the ethics approval gained for the study.

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
