# Peer review of "Mediating Effects of Rehabilitation Motivation between Social Support and Health-Related Quality of Life among Patients with Stroke"

_ijerph, 2022, doi:10.3390/ijerph192215274_

Round 1

Reviewer 1 Report

This manuscript represents the cross-sectional study aimed to identify the mediating effects of rehabilitation motivation on the relationship between social support and health-related quality of life among patients. Indeed, developing intervention strategies to motivate rehabilitation could improve outcomes in terms of HRQoL of patients after stroke. These are my comments and suggestions:

Abstract:

Abstract is clear and nicely written. However, I suggest to include quantitative results data.

Introduction:

Line 46: there is a typo here: "A meta-analysisi"

Line 95: there is grammatical error in formulation of hypotheses: "Social support is association" - it should be associated?

The rest of the Introduction chapter is nicely written, with adequate theoretical background, previous research on the topic and clear aim of the study.

Methods:

Please include the name of the scale used to measure social support. Did you register the use of EQ-5D? If yes, please mention it. Did you check for normality of data?

Results:

Please, avoid repeating same things in both text and tables.

Discussion and Conclusion:

Discussion is adequate. It includes clinical relevance of the study, future research proposals and comparison with previous studies. I have no further comments regarding these two chapters.

Reviewer 2 Report

1. The topic seems interesting and it is topical worldwide, from my point of view, the title is well-defined and matches the contents.
2. I recommend introducing the hypotheses to Materials and methods.
3. The introduction is generally well written, but in my opinion it would be good for your work to mention more citations to strengthen the rationale of your study, they are quite few.

Reviewer 3 Report

First of all, congratulations for the work carried out, it is a very interesting research and the methodology of the study is very well defined and developed. After reviewing the manuscript, it could be improved in the following points:

The introduction could be improved. Are there similar previous studies in other hospitals? Are there similar previous studies in other countries? The object of the study should be contextualized.

I think it would be interesting to include a "strengths of the study" section in the manuscript.

Best regards
